# Ultrasound in Sepsis and Septic Shock—From Diagnosis to Treatment

**DOI:** 10.3390/jcm12031185

**Published:** 2023-02-02

**Authors:** Gianluca Tullo, Marcello Candelli, Irene Gasparrini, Sara Micci, Francesco Franceschi

**Affiliations:** Emergency Medicine Department, Fondazione Policlinico Universitario Agostino Gemelli—IRCCS, Università Cattolica del Sacro Cuore di Roma, 00168 Rome, Italy

**Keywords:** sepsis, septic shock, ultrasound, fluid resuscitation, emergency medicine, critical care

## Abstract

Sepsis and septic shock are among the leading causes of in-hospital mortality worldwide, causing a considerable burden for healthcare. The early identification of sepsis as well as the individuation of the septic focus is pivotal, followed by the prompt initiation of antibiotic therapy, appropriate source control as well as adequate hemodynamic resuscitation. For years now, both emergency department (ED) doctors and intensivists have used ultrasound as an adjunctive tool for the correct diagnosis and treatment of these patients. Our aim was to better understand the state-of-the art role of ultrasound in the diagnosis and treatment of sepsis and septic shock. Methods: We conducted an extensive literature search about the topic and reported on the data from the most significant papers over the last 20 years. Results: We divided each article by topic and exposed the results accordingly, identifying four main aspects: sepsis diagnosis, source control and procedure, fluid resuscitation and hemodynamic optimization, and echocardiography in septic cardiomyopathy. Conclusion: The use of ultrasound throughout the process of the diagnosis and treatment of sepsis and septic shock provides the clinician with an adjunctive tool to better characterize patients and ensure early, aggressive, as well as individualized therapy, when needed. More data are needed to conclude that the use of ultrasound might improve survival in this subset of patients.

## 1. Introduction

Sepsis and septic shock are potentially fatal diseases and are among the leading causes of in-hospital mortality worldwide [1]. As widely demonstrated in the current literature, early recognition and timely treatment are critical to ensure better patient survival. The recognition of sepsis is challenging, especially in the emergency department (ED), and diagnostic and therapeutic flowcharts are far less standardized and easy to use compared with other common, time-dependent diseases. The proper identification of the septic patient and early treatment with antibiotics and source control are often delayed, and the diagnosis and immediate resuscitation of septic shock based only on traditional measures, such as clinical presentation and laboratory data, may take too much time to ensure a correct diagnosis and appropriate aggressiveness, according to current guidelines. In addition, appropriate fluid resuscitation measures and vasoactive medication administration can be challenging, as adequate fluid resuscitation and proper hemodynamic optimization are often complex and require individualized therapy, which can be particularly difficult in a crowded ED. In this situation, the use of point-of-care ultrasound (POCUS) has proven to be an excellent tool for emergency physicians and intensivists to treat septic patients. History taking and physical examination remain the fundamental elements in the clinical evaluation of all patients, especially in the setting of emergency and critical care when the use of laboratory analysis can be misguiding and might require too much time to obtain results. In this context, the use of point-of-care ultrasound when guided by a clinical suspicion can ensure correct and rapid diagnosis and treatment.

Ultrasound can improve the physician’s ability to correctly identify the focus of infection in patients with possible sepsis. In addition, the use of ultrasound as an aid in bedside procedures such as central venous catheter placement or percutaneous drainage is now the standard of care. Even in patients with undifferentiated shock [2], a rapid ultrasound assessment of cardiac function and volemic status can help clinicians understand the physiology and thus identify the possible septic cause of shock. Indeed, several protocols have been developed over the years in which ultrasound plays an important role in the management of critically ill septic patients in both the emergency department and the intensive care unit. In this narrative review, we discuss in detail the role of ultrasound in the diagnosis and management of patients with sepsis and septic shock.

## 2. Materials and Methods

We searched the medical literature from the past 20 years in the following electronic databases: MEDLINE/PubMed, Scopus, and Embase. The search terms were:-Sepsis AND ultrasound-Sepsis AND point-of-care ultrasound-Ultrasound AND septic shock-Ultrasound AND critical care-Ultrasound AND emergency department-Ultrasound AND procedures-Ultrasound AND hemodynamics

The search strategy was limited to articles that contained at least abstracts in English. The preferred studies were randomized clinical trials, followed by observational studies (both retrospective and prospective), or systematic reviews and meta-analysis. The articles were screened for search terms in the titles and abstracts. The studies were selected independently by two reviewers (I.G. and G.T.). An additional manual review of the references of the retrieved articles was performed to identify potentially relevant studies. We hereby present the results of our research in the form of a narrative review.

## 3. Results

We identified a total of 150 studies that met our prespecified criteria. Through a secondary analysis, we obtained a total of 70 studies that were included in this paper based on their relevance to the research topic. Among these studies, we identified 13 randomized clinical trials, 8 observational studies, 32 systematic reviews, and 5 meta-analyses (Table 1). In addition, we report on the data derived from five international guidelines, three international consensus documents, and a number of original research and cohort studies. After a careful analysis of the identified studies, we subdivided each paper according to the topic addressed and therefore present the following results with the main themes of sepsis diagnosis, source control and procedures, fluid resuscitation and hemodynamic optimization in septic shock, and echocardiography in sepsis cardiomyopathy.

### 3.1. Sepsis Diagnosis

Sepsis is defined as a life-threatening organ dysfunction caused by a dysregulated host response to infection [3]. The recognition of sepsis can be challenging. It requires an accurate history taking, physical examination and interpretation of laboratory data. As an aid to clinicians, several sepsis screening tools have been developed over the years to help correctly identify patients with sepsis. The most commonly used and standardized are the qSOFA (Quick Sequential Organ Failure Score), SIRS (Systemic Inflammatory Response Syndrome), and NEWS (National Early Warning Score) [4]. However, these scores are quite nonspecific. For example, the qSOFA quickly identifies the signs of organ dysfunction without considering a possible infectious cause, whereas the SIRS criteria identify patients who have the signs of a systemic inflammatory response without the obvious signs of organ dysfunction or an infectious cause for the inflammation. Therefore, clinical assessment, including history taking and a physical examination, remains the central element for the appropriate recognition of sepsis, as stated in the most recent international sepsis guidelines [3]. Nevertheless, this process can be supported by a number of tools, such as laboratory parameters [5], biomarkers [6,7], and point-of-care ultrasound [8]. One of the fundamental elements in the diagnosis of sepsis is the recognition and correct identification of the septic focus. According to the prevailing demographic data, the most common primary sources of infection in patients with sepsis are respiratory tract, urinary tract, and intra-abdominal sources, followed by less common causes such as skin and soft tissue infections, meningitis, and infections associated with indwelling catheters.

Respiratory tract. A large systematic review and meta-analysis conducted several years ago concluded that lung ultrasonography, when performed by trained personnel, performed incredibly well in the diagnosis of pneumonia: the pooled sensitivity and specificity for the diagnosis of pneumonia using LUS were 94% (95% CI, 92–96%) and 96% (94–97%), respectively [9]. In addition, another large meta-analysis examined the role of lung ultrasound in the evaluation of ED patients with undifferentiated dyspnea to correctly identify the underlying cause and make a differential diagnosis between pneumonia, COPD/asthma exacerbation, and heart failure [10]. On this particular topic, several ultrasound protocols have become established as common practice for the diagnostic evaluation of patients with dyspnea in the ED and ICU, such as the BLUE protocol [11]. Furthermore, acute respiratory distress syndrome (ARDS) is a common and feared complication of sepsis and septic shock, regardless of the primary source of infection [12]. Ultrasonography of the lungs in critically ill septic patients with respiratory distress could help identify this dangerous complication and guide the physician to more aggressive measures to treat respiratory failure, such as high-flow oxygen delivery or mechanical ventilation [13].

Urinary tract. Urinary tract infections are the second most common source of infection in sepsis. While the clinical presentation may be sufficient for the emergency physician to make a correct diagnosis UTI, ultrasound is the first choice for the evaluation of the kidneys and excretory organs because it is widely available and relatively simple and rapid for the evaluation of hydronephrosis and renal abscesses [14].

Abdominal sources. Abdominal infections are a relatively common cause of sepsis, and unlike the previously mentioned organs and systems, the clinical evaluation of the abdomen is complex and often misleading [15]. Complementary to patient history, clinical, and laboratory data, abdominal ultrasound is a widely available and relatively inexpensive examination that can be performed at the bedside and, when carried out by an experienced ultrasonographer, can identify various abdominal pathologies that may be the cause of infection, such as cholecystitis, cholangitis, pancreatitis, small bowel obstruction, or perforation [16]. Ultrasound is the imaging modality of choice for diagnosis in patients with suspected acute cholangitis or gallbladder disease, a common and very serious cause of abdominal infection that should be treated promptly [17]. As for appendicitis, it is generally accepted that the clinical findings in conjunction with an ultrasound are sufficient for the diagnosis of acute appendicitis, whereas a CT examination should be reserved for patients with inconclusive sonographic findings [18,19].

### 3.2. Source Control and Procedures

Source control. The use of ultrasound as an aid in performing diagnostic and interventional procedures has become the standard-of-care. Thoracentesis is a commonly performed procedure in the ED and in the intensive care unit (ICU), for both diagnostic and therapeutic purposes. Ultrasound is known to be a fundamental tool to support the procedure, increasing safety and reducing the risk of a potentially fatal complication [20,21]. In addition, it allows for identification of the characteristics of pleural effusion and differentiation of the type of effusion based on the different pattern of echogenicity or the presence of septa or obvious empyema. As for the abdomen, several procedures may be required to identify the septic focus and subsequently control the source. In patients with ascites, performing a diagnostic and/or evacuative paracentesis is crucial, especially if spontaneous bacterial peritonitis is suspected [22,23,24]. In the context of biliary tract and gallbladder disease, ultrasound is the imaging modality of choice because the anatomic location is easily accessible, and the good echo pattern of the biliary systems allows a correct diagnosis. In patients with sepsis due to ascending cholangitis or with cholecystic empyema in whom the surgical risks are unacceptable, ultrasound-guided drainage offers a good alternative to the standard procedure of endoscopic retrograde cholangiopancreatography (ERCP) to control the septic focus, especially if the patient is hemodynamically unstable [25]. In the case of obstructive uropathy, the most common treatment is to either place a ureteral stent or perform a percutaneous nephrostomy to decompress the renal system and prevent the occurrence of renal failure or infectious complications [26]. The urgent placement of a percutaneous drainage tool is the most commonly performed treatment in the presence of sepsis, both under ultrasound guidance and fluoroscopy [27]. As mentioned earlier, after respiratory, urinary, and abdominal infections, skin, soft tissue, and central nervous system infections are the most common causes of sepsis. While the diagnosis of skin and soft tissue infections can be made by history taking and physical examination, ultrasound could be an additional tool to better characterize the extent of infection and identify abscesses [28]. Even when the clinical identification of soft tissue infection is straightforward, distinguishing between simple cellulitis and the presence of a subcutaneous abscess can be challenging. In this case, the use of ultrasound can help the clinician correctly identify the extent of the pathology, as described in a recent systematic review [29]. Meningitis is a relatively rare but potentially fatal condition that occurs in Eds and ICUs. The most important technique for identifying bacterial and viral meningitis is a lumbar puncture [30]. Although there are no international guidelines recommending the use of ultrasound for the procedure [31], some position papers recommend the use of ultrasound to correctly identify the puncture site, especially in patients with a difficult anatomy, to reduce the number of needle insertions and increase the overall success rate [32].

Ultrasound in bedside procedures. As widely recognized in the scientific literature and international guidelines, the management of patients with shock includes a range of noninvasive and invasive procedures for advanced multiparametric monitoring and hemodynamic assessment [33]. In this regard, the recent guidelines from the Surviving Sepsis Campaign recommend that central venous catheters for vasoactive drug infusion and invasive arterial monitoring be placed as soon as possible in patients with septic shock [3]. The current scientific literature suggests that ultrasound guidance during central venous catheter placement [34], including in the setting of the ED [35], increases the likelihood of a successful first attempt and significantly decreases the rate of complications, such as accidental arterial puncture, hematoma formation, and pneumothorax, especially during a puncture of the internal jugular vein [36], compared with a standard procedure based on landmarks [37]. In addition, the correct positioning of a central venous catheter can be confirmed by ultrasound by visualizing microbubble artifacts in the right atrium after the injection of saline through the distal port. The so-called “bubble test” has been shown to be much faster than the conventional confirmation method (a standard antero-posterior chest radiograph) and to expedite the use of CVCs in critically ill patients in the ED [38]. The most common cannulation site for arterial lines is the radial artery, followed by the femoral or brachial artery [39,40]. Although the traditional method for catheterization of the radial artery is landmark-guided palpation, the use of ultrasound not only increases the first-attempt success rate but also significantly reduces the complication rate and the number of failed attempts, as shown in a randomized controlled trial [41].

### 3.3. Fluid Resuscitation and Hemodynamic Optimization in Septic Shock

As outlined in the most recent guidelines for the management of patients with septic shock, the mainstay of treatment, in addition to antimicrobial therapy and control of the source of infection, is appropriate hemodynamic optimization and the restoration of adequate tissue perfusion and oxygenation. The standard treatment for patients with sepsis and hypotension or a rise in lactate >4 mmol/L is to initiate fluid resuscitation with approximately 30 mL/kg of balanced crystalloids within three hours. After the initial bolus, the successive administration of fluid should be tailored to dynamic measures of fluid responsiveness, which can be achieved by a variety of invasive and noninvasive methods [33]. In the meantime, if hypotension persists, timely vasopressor therapy, preferably with norepinephrine, should be initiated, with vasopressor and/or inotropic therapy adjusted according to the measures of cardiac output and tissue perfusion. Recent literature has debated whether a patient-tailored approach to fluid resuscitation and vasopressor/inotropic support in patients with septic shock leads to better outcomes than standardized treatment as suggested by the SSC guidelines. This strategy of early targeted therapy was first proposed in a cornerstone study published in 2001 by Rivers et al. in which the authors proposed an algorithm based on hemodynamic optimization through targeted resuscitation using central venous pressure (CVP), mean arterial pressure (MAP), and central venous oxygen saturation (ScvO2) [42]. Although the results of this study have not been confirmed by subsequent research and have been controversial over the years, the concept behind the algorithm proposed by Rivers et al. is still valid today.

Fluid management. One of the main focuses of current research continues to be determining the correct amount of fluid for resuscitation. With regard to fluid management, some authors suggest that liberal fluid resuscitation strategies may harm patients by increasing the risk of respiratory failure due to ARDS [43] and all-cause mortality [44]. However, a recently published large study involving more than 1500 participating patients showed that restricting intravenous fluid intake after an initial bolus of 1000 mL in patients admitted to the ICU with septic shock did not result in lower mortality within 90 days compared with standard intravenous fluid therapy [45]. A similar result was provided by another study in which the patients were randomized to a restrictive fluid strategy, defined as <60 mL/kg crystalloids in the first 72 h after the first bolus of 1000 mL, versus standard care. The patients treated with the restrictive strategy received significantly less fluid (47.1 vs. 61.1 mL/kg; *p* = 0.01), but there was no difference between the two groups in mortality, new organ failure, length of hospital or intensive care stay, or serious adverse events at 30 days [46]. Therefore, several ultrasound-guided protocols have been developed over the years to guide fluid resuscitation according to fluid responsiveness, with the aim of helping clinicians choose the right approach to balance hemodynamics and the risk of excessive fluid intake. Yongyuan et al. demonstrated that fluid resuscitation guided by a pulse index and continuous cardiac output (PICCO) monitoring combined with transabdominal ultrasound resulted in increased lactic acid clearance as well as shorter duration of mechanical ventilation, lower total fluid intake, and a significantly shorter length of hospital stay [47]. In another study, patients were randomly divided into two groups, either a group receiving integrated cardiovascular ultrasound (ICUS) within 1 h of admission for hemodynamic decision-making or a control group that received standard care [48]. There was no significant difference in mortality, but the patients treated with the ICUS protocol received more fluid in the first 6 h and a shorter total duration of vasopressor support. Interestingly, another study addressed this issue and examined the efficacy of dynamic measures of fluid responsiveness in septic shock to guide fluid resuscitation and improve patient outcomes as compared to usual care [49]. Fluid responsiveness was defined as a significant (>10%) increase in stroke volume after a passive leg raise test. The patients treated according to the study arm of the protocol had significantly lower fluid balance compared with the control group (−1.37 L in favor of the intervention arm; 0.65 ± 2.85 L intervention arm vs. 2.02 ± 3.44 L usual care arm; *p* = 0.021) and fewer patients required renal replacement therapy (5.1% vs. 17.5%; *p* = 0.04) or mechanical ventilation (17.7% vs. 34.1%; *p* = 0.04). In this study, stroke volume was measured by a noninvasive bioreactance analysis. As we know, the transthoracic echocardiographic (TTE) evaluation of LVOT VTI is a potential surrogate measure of stroke volume and cardiac output. In fact, this method was used to estimate the variation in stroke volume after passive leg raising in a clinical trial to guide fluid resuscitation [50]. The result was that patients treated with the strategies guided by passive leg raising and TTE stroke volume calculation had improved perfusion and oxygenation of tissues and organs, a reduced incidence of pulmonary edema caused by rapid fluid administration, and a shortened hospital stay, but there was no significant effect on hospital mortality.

Fluid tolerance and vasopressor support. Although the appropriate response to fluid resuscitation is tailored to increase cardiac output, it is still important to assess the limits of fluid resuscitation using the parameters of fluid tolerance. In this case, it is clear that clinical assessment alone is not always sufficient to detect the signs of significant fluid congestion. Therefore, several ultrasound-based protocols have been developed over the years that take into account both the fluid tolerance and fluid responsiveness of the patient by incorporating ultrasonography of the lungs into the assessment [51,52]. In particular, the appearance of B-lines in the lungs may be an early sign of pulmonary congestion and prompt the physician to reduce fluid intake. However, in a recent randomized clinical trial of ultrasound-guided fluid resuscitation for sepsis, the authors did not demonstrate an increase in survival when the fluid administration was based on the ultrasound assessment of inferior vena cava diameter. Interestingly, however, the ultrasound-guided protocol resulted in a lower total volume of fluid administered [53]. A similar conclusion was reached in another clinical trial, in which the authors found that resuscitation based on an early echocardiographic assessment of hemodynamics compared with the standard SSC care had significant differences in the amount of fluids administered and in the early initiation of inotropic support [54]; specifically, the echocardiography group received fewer fluids than the standard care group and inotropic support was initiated earlier than according to SSC guidelines. Nevertheless, a large meta-analysis concluded that resuscitation guided by volume-based measures does not reduce mortality in sepsis [55]. A recent prospective cohort study [56] proposed a promising protocol in which a carotid Doppler wave analysis as a measure of fluid responsiveness and a radial artery resistance index as a marker of systemic vascular resistance could help clinicians appropriately administer fluids and initiate vasopressor support. The patients treated according to the proposed protocol received fewer fluids, were treated earlier with vasopressors, and had a shorter hospital stay, but no significant difference in mortality was observed compared with standard care. However, it is clear that the hemodynamic phenotypes of septic shock are complex and often interwoven with multiple concurrent pathophysiologic disturbances [57,58]. In this context, the use of ultrasound as an early tool to characterize the patient’s hemodynamic profile and thus to dynamically adjust the therapeutic approach seems to be a promising strategy [59,60]. Table 2 summarizes the most important clinical research in recent years related to ultrasound-guided fluid management in septic shock.

### 3.4. Echocardiography in Sepsis Cardiomyopathy

Sepsis-induced cardiomyopathy (SIC) is generally and increasingly recognized as a potentially dangerous complication of sepsis and septic shock [61]. According to various reports in the literature, the prevalence of cardiomyopathy in sepsis and septic shock ranges from 18 to 40% and in some reports is as high as 70% [62], probably reflecting the variability of its definition and recognition in the clinical setting. To complicate matters, the impact of SIC on prognosis is not clear. While some authors have suggested that the presence of SIC increases mortality in patients with sepsis [63], others have found that the classic functional parameters such as LV ejection fraction (LVEF) do not correlate with mortality. This may be due to the inherent inadequacy of these parameters as measures of LV performance in the context of complex hemodynamic disorders [64,65]. In a retrospective cohort study, Sato et al. examined the epidemiology and clinical features of SIC in a cohort of patients with sepsis or septic shock [65]. The authors defined SIC as an LVEF < of 50% and an EF decrease from baseline ≥10% that recovered within 2 weeks of the acute event. The overall prevalence of SIC was 13.8%, with a significantly higher prevalence in men than in women (*p* = 0.02). Multivariate logistic regression analyses revealed that incidence was associated with younger age, a higher lactate level at admission, and a history of heart failure (HF). Nevertheless, there were no significant differences in in-hospital and 30-day mortality between the patients with and without sepsis-related cardiomyopathy, whereas the length of hospital and intensive care stay was significantly longer in the patients with SIC than in the patients without SIC. A similar conclusion was reached in a large meta-analysis that examined the prognostic significance of myocardial dysfunction in patients with sepsis and septic shock [66]. The authors concluded that the presence of new LV systolic dysfunction associated with sepsis, defined as low LVEF, was neither a sensitive nor a specific predictor of mortality. In contrast to these data, other authors have recently found that SIC, defined as a significant decrease in LVEF compared with an echocardiogram performed within 12 months, estimated by focused cardiac ultrasound performed by an emergency physician in the ED, was significantly associated with mortality at 90 days (OR 6.1) [67]. It must be emphasized that several echocardiographic parameters, such as LVEF, depend on the loading conditions of the cardiovascular system (i.e., preload and afterload). Therefore, it may be difficult to distinguish when the observed dysfunction is primarily myocardial and when it is caused by the hemodynamic disturbances that occur in the setting of septic shock. Over the years, other integrated echocardiographic measures of LV function have been developed that take into account the loading conditions of the cardiovascular system, such as afterload-dependent cardiac output or ventriculo-arterial coupling [68], but the discussion of these fascinating topics is beyond the scope of this review. In general, sepsis-related cardiomyopathy can be recognized as acutely decreased left ventricular (LV) contractility, possibly accompanied by LV dilatation with or without right ventricular failure, as observed on transthoracic ultrasound. This wide range of ultrasound findings reflects both a lack of consensus in the SIC definition and the underlying complexity of the pathophysiology and hemodynamic profile. It has been suggested that there are multiple hemodynamic clusters or cardiovascular phenotypes of septic shock. This cardiovascular profiling not only provides a clearer picture of the hemodynamic disturbances that occur in these critically ill patients, but also presumably allows prognostic stratification by identifying patients at increased risk of mortality, and could help clinicians appropriately optimize hemodynamic support with fluid resuscitation, vasopressors, and inotropic support [69,70].

## 4. Conclusions

An ultrasound-guided approach to sepsis and septic shock seems to improve the clinician’s ability to correctly manage this complex pathology. It is an additional tool available to the emergency physician in addition to the clinical history and physical examination, which remain essential for an appropriate diagnosis of sepsis and septic shock. Both the diagnosis and treatment of these conditions needs to be tailored to the patient, and ultrasound seems a promising tool in this difficult task. In particular, ultrasound-guided individualized hemodynamic optimization showed interesting results in the recent literature, but more data are needed to better understand if this strategy might significantly reduce mortality in sepsis and septic shock.

## Figures and Tables

**Table 1 jcm-12-01185-t001:** This table lists the types of articles which are included in the present review.

Type	No
Total studies	70
Review	32
Randomized clinical trials	13
Observational studies (prospective and retrospective)	8
Meta-analysis	5
International guidelines	5
Consensus documents	5
Others (cohort studies, editorials)	4

**Table 2 jcm-12-01185-t002:** This table summarizes the most significant clinical trials which have addressed ultrasound-guided fluid management in septic shock in recent years.

Authors	Title	Type	Technique	Findings
Li et al. [50]	Clinical value of early liquid resuscitation guided by passive leg-raising test combined with transthoracic echocardiography in patients with septic shock (article in chinese)	RCT, monocentric	PLR + SV variation through TTE	Early fluid resuscitation treatment strategies guided by the PLR combined with the TTE calculation of stroke volume could better improve the perfusion and oxygenation level of tissues, avoid pulmonary edema, and shorten the duration of hospitalization, but had no significant effect on hospital mortality.
Musikatavorn et al. [53]	Randomized Controlled Trial of Ultrasound-guided Fluid Resuscitation of Sepsis-Induced Hypoperfusion and Septic Shock	RCT, monocentric	respiratory change in IVC diameter	Ultrasound-guided resuscitation did not result in improved 30-day survival with respect to standard care; however, the US guided approach resulted in less fluids administered.
Li et al. [48]	Effect of focused cardiopulmonary ultrasonography on clinical outcome of septic shock: a randomized study	RCT, monocentric	ICUS vs. standard care	Patients treated with ICUS-guided fluid resuscitation received more fluids in the first 6 h, and the duration of vasopressor support was shorter with respect to the control group. No difference in mortality at 28 days.
Yao et al. [47]	Clinical Application of Transabdominal Ultrasound Combined with PICCO in Septic Shock Fluid Resuscitation and Its Predictive Value for Survival Outcome	RCT, monocentric	transabdominal ultrasound + PICCO	The combination of transabdominal ultrasound with PICCO is better at guiding fluid resuscitation in patients with septic shock with respect to PICCO alone; the experimental group had reduced BLA, mechanical ventilation time, total fluid input, and duration of hospitalization.
Devia Jaramillo et al. [56]	USER Protocol as a Guide to Resuscitation of the Patient with Septic Shock in the Emergency Department	Prospective, cohort study	PLRcD test + DSIR	The USER protocol showed lower fluid balances at 4 and 6 h, and a reduction in the total duration of hospitalization, as well as earlier initiation of norepinephrine and a statistically significant faster improvement in blood pressure. There were a trends in the reduction of mortality, and the length of ICU stay and total duration of hospitalization, although this was not significant.

BLA: blood lactic acid; DSIR: doppler snuffbox resistance index; ICU: intensive care unit; ICUS: integrated cardiopulmonary ultrasound; IVC: inferior vena cava; PICCO: pulse index continuous cardiac output; PLR passive leg raising; PLRcD: passive leg raising guided by carotid doppler; RCT: randomized clinical trial; SV: stroke volume; TTE: trans-thoracic ultrasound; US: ultrasound.

## Data Availability

Not applicable.

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
