# Peer review of "Ultrasound in Sepsis and Septic Shock—From Diagnosis to Treatment"

_jcm, 2023, doi:10.3390/jcm12031185_

Round 1
Reviewer 1 Report
This is an extensive and well written review on the use and benefits of ultrasound in the management of patients with sepsis and septic shock. Despite of many positive aspects, I have several concerns regarding this manuscript.
Major Comments:
1. The information summarized in this review is not new or newly synthesized. Similar efforts have been made before. For example: Shresta GS, et al. Rev Recent Clin Trials 2018; 13:243; Sweeney DA, et al. Semin Respir Crit Care Med 2021; 42:641.
2. The manuscript does not follow the PRISMA recommendations. Moreover, it is unclear what kind of review this is. On the first sight, it appears to be a systematic scoping review but when reading the manuscript, it becomes clear that this is a narrative review. No meta-analysis has been performed although binary outcome variables could have been systematically assessed.
3. Was the protocol for this review registered or published?
4. Although I am an absolute fan and regular user of point-of-care ultrasound for the management of acutely and critically ill patients (including those with sepsis), the manuscript overrates the importance of ultrasound (even) from my point of view. I suggest that the authors better highlight that ultrasound is ONE important diagnostic tool IN ADDITION to several other valuable tools (e.g., history taking, clinical examination, laboratory assessment) to manage patients with sepsis and septic shock.
5. The manuscript is long and contains only few sub-headings making it difficult to read. Some parts (e.g., the Discussion section) could be entirely omitted.
6. I am not sure what the message of Table 2 should be. While some of the presented studies evaluated the use of ultrasound as their primary study endpoint others did not.
Minor Comments:
1. Table 1 is an insufficient modification of the PRISMA flow diagram.
2. Please consistently use one term for ultrasound (ultrasound vs. sonography vs. echography).
3. Page 3, lines 108/109: “Even though chest infection is not the main cause of sepsis, …” contradicts the statement on the same page, lines 95/96 that “…, the most common primary sources of infection in patients with sepsis are respiratory tract, …”.
4. Please avoid superlative adjectives such as “performed incredibly well” and “very serious cause of abdominal infection” (all page 3).
5. Page 4: The first lines repeat parts of the paragraph “3.1. Sepsis Diagnosis”.
6. Page 5, line 217: What is meant with “by increasing the risk of respiratory arrest due to ARDS”. The major risk associated with ARDS is acute respiratory insufficiency (with all pathphysiological consequences) but not primarily respiratory arrest.
7. Page 5, lines 231-234: What was the comparator in this study? Please indicate.
8. The heading of Table 2 is insufficient.
Author Response
Reviewer 1
Comments and Suggestions for Authors
This is an extensive and well written review on the use and benefits of ultrasound in the management of patients with sepsis and septic shock. Despite of many positive aspects, I have several concerns regarding this manuscript.
We would like to thank the reviewer for taking the time to evaluate our study, and for the valuable comments that allowed us to improve the paper
Major Comments:
- The information summarized in this review is not new or newly synthesized. Similar efforts have been made before. For example: Shresta GS, et al. Rev Recent Clin Trials 2018; 13:243; Sweeney DA, et al. Semin Respir Crit Care Med 2021; 42:641.
Thanks for the suggestion. We are aware that the topic is not new, and many interesting papers are present on literature that addressed the same issues. Nonetheless, the structure of our review is different, and to some extent more extensive than the interesting papers you cited. We dedicated a section to fluid management hemodynamic optimization in septic shock and thoroughly analyzed this complicated issue while the other authors focused on the role of ultrasound in the approach to undifferentiated shock; furthermore, we stressed the topic of echocardiography in sepsis-induced cardiomyopathy.
- The manuscript does not follow the PRISMA recommendations. Moreover, it is unclear what kind of review this is. On the first sight, it appears to be a systematic scoping review but when reading the manuscript, it becomes clear that this is a narrative review. No meta-analysis has been performed although binary outcome variables could have been systematically assessed.
Our review is a narrative one therefore we did not strictly apply PRISMA recommendations. We proceeded to specify this feature in the materials and methods section.
- Was the protocol for this review registered or published?
No, we did not register nor publish the protocol for this review.
- Although I am an absolute fan and regular user of point-of-care ultrasound for the management of acutely and critically ill patients (including those with sepsis), the manuscript overrates the importance of ultrasound (even) from my point of view. I suggest that the authors better highlight that ultrasound is ONE important diagnostic tool IN ADDITION to several other valuable tools (e.g., history taking, clinical examination, laboratory assessment) to manage patients with sepsis and septic shock.
Thanks for the suggestion. We already stressed the fact that ultrasound is an adjunctive tool in the clinical evaluation and management of septic patients, apart from clinical evaluation and laboratory assessment which remain the pivotal elements for the ED physician. Nonetheless, we will stress the topic more.
- The manuscript is long and contains only few sub-headings making it difficult to read. Some parts (e.g., the Discussion section) could be entirely omitted.
Thanks for the suggestion. The length of the manuscript complies to the rules of the journal to produce at least 4000 words. Nonetheless, we divided each segment into more sub-headings in order to ease the reading.
- I am not sure what the message of Table 2 should be. While some of the presented studies evaluated the use of ultrasound as their primary study endpoint others did not.
Thanks for the suggestion. The message of Table 2 should be to address the difficult topic of fluid management in septic shock; we decided to include trials who did not use ultrasound as the primary endpoint in order to enrich the discussion on the topic and also because, unfortunately, there are not many good-quality clinical trials on the use of ultrasound in fluid management in the setting of septic shock at present in literature.
Minor Comments:
- Table 1 is an insufficient modification of the PRISMA flow diagram.
Table 2 was not intended as a variation of the PRISMA rules, but as a simple specification of the type and number of papers included in our narrative review
- Please consistently use one term for ultrasound (ultrasound vs. sonography vs. echography).
Thanks for the suggestion, we modified the manuscript to provide a uniform lexicon.
- Page 3, lines 108/109: “Even though chest infection is not the main cause of sepsis, …” contradicts the statement on the same page, lines 95/96 that “…, the most common primary sources of infection in patients with sepsis are respiratory tract, …”.
Thanks for this suggestion. The phrase was unclear: ARDS can be a serious complication of sepsis even when the primary source of infection of the septic patient is not pulmonary. We rephrased the sentence in order for it to be more fluent.
- Please avoid superlative adjectives such as “performed incredibly well” and “very serious cause of abdominal infection” (all page 3).
Thanks for the suggestion.
- Page 4: The first lines repeat parts of the paragraph “3.1. Sepsis Diagnosis”.
Thanks for the suggestion, we modified the paper accordingly.
- Page 5, line 217: What is meant with “by increasing the risk of respiratory arrest due to ARDS”. The major risk associated with ARDS is acute respiratory insufficiency (with all pathphysiological consequences) but not primarily respiratory arrest.
Thanks for the suggestion, it was a mistake.
- Page 5, lines 231-234: What was the comparator in this study? Please indicate.
The comparator in the study was “usual care”. Thank for the suggestion, we modified the paper accordingly.
- The heading of Table 2 is insufficient.
Thanks for the suggestion, we added the heading which was missing.

Reviewer 2 Report
It is a well describe article related to ultrasound in sepsis and septic shock.
Good summary compare between other modality in assess hemodynamic management and using ultrasound. It is a powerful tool diagnostic and help in managament.
May consider adding Echocardiogram and sepsis in table summary
Author Response
REVIEWER 2
Comments and Suggestions for Authors
It is a well describe article related to ultrasound in sepsis and septic shock.
Good summary compare between other modality in assess hemodynamic management and using ultrasound. It is a powerful tool diagnostic and help in managament. May consider adding Echocardiogram and sepsis in table summary
We would like to thank the reviewer for his appreciation. We have made some changes, which you will find in red letters in the text. I hope you like these changes

Round 2
Reviewer 1 Report
The authors have submitted a revised version. The work has partially improved. The fact that previous reviews have addressed the topic, that the review is not systematic according to PRISMA recommendations and that the protocol was not pre-published are limitations impossible to change. Other relevant concerns remain, as some of them have only been addressed as a response to the reviewer but not in the revised manuscript itself.
1. I could not identify further statements or discussion of the fact that the ultrasound is an important additional tool for the management of patients with sepsis and septic shock in the revised manuscript, at least not highlighted. Please stress this aspect more, as indicated in your response.
2. Thank you very much for including sub-headings. In my opinion, this has improved the flow of the manuscript a lot. Although the authors remain within the recommended journal word limit for reviews, I am unsure what additional information the Discussion section adds for the reader. Therefore, I suggest to omit this paragraph from the manuscript.
3. I do not agree that it is meaningful to include non-ultrasound-related publications into Table 2, as the selection of cited references becomes highly arbitrary. Please include only publications related to the use of ultrasound and focus on advantages and disadvantages.
Author Response
The authors have submitted a revised version. The work has partially improved. The fact that previous reviews have addressed the topic, that the review is not systematic according to PRISMA recommendations and that the protocol was not pre-published are limitations impossible to change. Other relevant concerns remain, as some of them have only been addressed as a response to the reviewer but not in the revised manuscript itself.
Answer: We would like to thank the reviewer again for taking the time to review our work. We also thank him for helping us to improve it significantly.
- I could not identify further statements or discussion of the fact that the ultrasound is an important additional tool for the management of patients with sepsis and septic shock in the revised manuscript, at least not highlighted. Please stress this aspect more, as indicated in your response.
Answer: We have included in both the introduction and conclusions the message that ultrasound is an additional tool in the hands of the emergency physician to assess sepsis and septic shock. We have emphasized the fundamental role of clinical history and physical examination, as suggested by the reviewer
- Thank you very much for including sub-headings. In my opinion, this has improved the flow of the manuscript a lot. Although the authors remain within the recommended journal word limit for reviews, I am unsure what additional information the Discussion section adds for the reader. Therefore, I suggest to omit this paragraph from the manuscript.
Answer: We have deleted the discussion as suggested by the reviewer
- I do not agree that it is meaningful to include non-ultrasound-related publications into Table 2, as the selection of cited references becomes highly arbitrary. Please include only publications related to the use of ultrasound and focus on advantages and disadvantages.
Answer: we have changed the table 2 as requested by the reviewer